# A Bibliometric Analysis to Identify Research Trends in Intervention Programs for Smartphone Addiction

**DOI:** 10.3390/ijerph20053840

**Published:** 2023-02-21

**Authors:** Yi-Ying Wu, Wen-Huei Chou

**Affiliations:** 1Graduate School of Design, National Yunlin University of Science and Technology, 123 University Road, Section 3, Douliou 64002, Yunlin, Taiwan; 2Department of Digital Media Design, National Yunlin University of Science and Technology, 123 University Road, Section 3, Douliou 64002, Yunlin, Taiwan

**Keywords:** smartphone addiction, research trend, intervention program, bibliometric analysis, LDA

## Abstract

Smartphone addiction is a serious social problem that necessitates research. To identify trends in intervention programs for smartphone addiction, distribution of research topics, and inter-relationships in academic research. We analyzed 104 studies published between 30 June 2022, and 31 August 2022, on the Web of Science (WoS). We applied the bibliometric method and identified the relationship and development trends of academic research in the field using descriptive analysis, the Latent Dirichlet Allocation (LDA), co-citation analysis, bibliographic coupling, and co-occurrence. There were four findings: First, intervention programs are classified into 10 types: psychological, social support, lifestyle, technological, family, medical care, educational, exercise, mindfulness, and meditation. Second, the volume of research on intervention programs increased every year. Third, China and South Korea had the highest research involvement. Finally, academic studies were split into either the human behavior or social science categories. Most of them defined the symptoms in terms of individual behavior and social relations, implying that smartphone addiction is not yet recognized as a disorder. Smartphone addiction has not yet been internationally recognized as a disorder, despite its impact on human physiology, psychology, and social behavior. Most related studies have been conducted in Asia, specifically in China and South Korea; Spain has the most outside Asia. Additionally, most of the research subjects were students, probably because of the convenience of sampling. As smartphones gain popularity among older adults, future studies could focus on smartphone addiction among individuals of varied ages.

## 1. Introduction

Smartphones are technological tools used by the majority of people in developed countries. They can be used for making transactions and investments, searching for information and geographical locations, taking and editing photos, learning, identifying callers, listening to music, checking the time, using social media and communication apps, integrating control systems for medical treatment, and as interventions for problems such as alcoholism, gambling, smoking, or smartphone addiction [1,2,3,4,5,6]. Thus, smartphones play a significant role in people’s daily lives [7]. A 2022 study suggested that China, Saudi Arabia, and Malaysia are the countries with the highest smartphone addiction rates [8]. Another 2022 study indicated that the current global prevalence rate of this addiction is estimated to be 26.99%, whereas the rate of social media addiction is 17.42%, internet addiction is 14.22%, online pornography addiction is 8.23%, and video game addiction is 6.04% [9]. Although an upward trend was noted even prior to the outbreak of COVID-19, smartphone addiction has increased after the pandemic [10].

The existing literature on smartphone addiction intervention emphasizes that problematic use can trigger severe psychological problems, including depression, anxiety, inattention, stress, and suicidality [11,12]; behavioral issues, such as dependence, attachment (e.g., anxious, avoidant attachment styles), and excessive gaming and social media use [13,14], as well as physical complications, including obesity, skeletal problems, and body posture [15,16]. Several studies have even explored personality factors, such as neuroticism [17].

Although the literature indicates that smartphone addiction is a serious problem, it is not internationally recognized as a disorder. Only internet gaming disorder (IGD) is classified by the World Health Organization (WHO) in the 11th revision of the International Classification of Diseases (ICD-11) [18] and the American Psychiatric Association (APA) in the 5th edition of the Diagnostic and Statistical Manual of Mental Disorders (DSM-5) [19]. Despite many studies suggesting that smartphone addiction is related to physical and mental symptoms, it is not internationally recognized as a disorder.

The advent of smartphones has had merits as well as demerits [20]. The prevalence rate of smartphone addiction has increased, consequently, affecting people’s lives and prompting the attention of interventional research. Hence, this study used bibliometric analysis and the Latent Dirichlet Allocation (LDA) to understand the development trends of smartphone addiction intervention programs, as well as relevant literature, journals, and researchers, and analyzed the underlying academic communities behind the literature on intervention programs, using network analysis.

The objective of this study was to understand the recent research trends of intervention programs for smartphone addiction and explore their future developments. Based on the above background, the study discusses intervention programs for smartphone addiction, focusing on the following research questions:*What is the classification of intervention programs for smartphone addiction and the knowledge structure behind them?**What are the growth trends, quantity, and regional distribution of research studies on intervention programs for smartphone addiction?**Which are the influential journals, authors, and studies on intervention programs for smartphone addiction?*

### 1.1. Intervention in Smartphone Addiction

The significance of smartphone addiction intervention lies in the suffering of a few people from physical and mental distress, which affects their daily life; however, following an intervention, patients can return to their previous lives. Currently, there is little consensus on any intervention program for smartphone addiction; however, there are a few influential studies regarding addiction intervention. To clarify, an intervention program is a facilitation process based on cooperation and trust; therefore, it is necessary to conduct in-depth research on the causes and treatments of smartphone addiction. However, studies have shown that traditional clinical methods using surveys and interviews have serious limitations, while medical professionals may not be able to sustain intervention in addiction-affected groups, which leads to questionable evaluations of subjective perceptions [21,22]. Moreover, previous, and current studies on smartphone addiction often focus on exploring and intervening in the individual’s physical and mental conditions and societal relationships. In particular, the most unusual intervention program at present is meditation intervention, which applies self-regulation of the mind and meditation to change personal habits [23].

Furthermore, a 2018 study on female adolescents found five effective interventions: involuntary restriction, self-control and self-awareness, school restrictions, peer support, and therapeutic professional services [24]. A literature search revealed that studies on intervention programs started to increase many years ago, with experimental research carried out. Through years of published research, the domain of smartphone addiction intervention gradually received attention.

### 1.2. Bibliometric Analysis

Bibliometric analysis may not be a novel research method, yet it is rigorous in exploring and analyzing scientific data. Owing to the popularity of journal research databases, this method has recently resurfaced. It enables us to understand the trend changes in a particular domain and illustrate emerging developments in that particular field [25]. It is mainly applied in the analysis of scientific research publications, thematic classifications, journal sources, geographic distribution, keyword usage, and the relationship between article topics and quantity calculations [26].

In particular, visualization analysis can derive information from numerous sources [27] to help scholars make decisions based on data. Visualization integrates various perspectives and provides descriptive and predictive analyses [28,29], including (1) citation analysis; (2) co-citation analysis; (3) bibliographic coupling; and (4) co-occurrence [25,30].

Let us discuss these analysis methods in bibliometrics. First, “citation analysis” refers to ranking journals according to the number of studies published and the total number of citations, to indicate the influence of journals [31]. Second, “co-citation analysis” is used as an effective method to identify the knowledge structure of a research domain, relying on simple co-citation counts to provide detailed information about sub-disciplines, and measuring the similarity between co-cited authors by comparing the authors’ citation content [32]. Third, “bibliographic coupling” refers to the determination of the correlation between two articles based on their common references and the analysis of scientific literature structures and cooperation between countries using the network of relationships between different elements [33,34]. Finally, “co-occurrence” is a keyword used to describe the subject area to determine the relationship between the main concepts involved in the research domain [35].

### 1.3. Latent Dirichlet Allocation (LDA)

There is a diverse range of topic modeling methods, of which the LDA is the most popular one in domain research [36]. Therefore, this study adopted LDA as an article classification method. Each document represents a fixed probability distribution of latent topics [37,38]. Topics are classified into conditions based on features such as “paragraph” or “sentence,” to provide a powerful text model that uses software analysis to obtain information [39], including the following: (1) topic modeling over various domains; (2) hierarchical topic modeling; (3) word-embedded topic models; (4) topic models in multilingual perspectives [40].

Moreover, topic modeling is a research method used for retrieval and text mining that infers a generative probabilistic model of text databases deduced from the data [41]. Each document is produced by a mixture of topics; thus, the continuous-valued mixture proportions are distributed as a LDA random variable [42]. This method can also observe words and calculate the distribution of words and sentences in latent topics [43].

Based on the above discussion, the advantages of LDA include its ability to estimate words that appear infrequently and use thesauruses and vocabulary to improve topic cohesion within and across languages; its disadvantage is that the model will vary over time, thereby requiring studies to run calculations repeatedly [44,45].

## 2. Method

This study applied bibliometrics while adopting descriptive analysis, the LDA, and network visualization analysis as key analysis methods. The objective was to understand the trend of interventions for smartphone addiction by analyzing the concepts of scale, time, space, and composition. As shown in Figure 1, the workflow started by collecting journal articles from June to August 2022, excluding irrelevant and incompletely published studies, and performing bibliometric analyses.

### 2.1. Data Source

This study explores the research domain of intervention programs for smartphone addiction by collecting literature from the Web of Science (WoS). The search process commenced on 30 June 2022, and the collection of journal articles ended on 31 August 2022.

### 2.2. Target Searching

The study focused on the following: (1) searching for studies on intervention programs for smartphone addiction published from 2014 to 2021, and the earliest intervention study after the screening was published in 2014; and (2) comparing the number of studies on intervention programs within those eight years. The study excluded (1) intervention studies on alcoholism, gambling, and smoking, where smartphones are involved; (2) studies on the use of smartphones in the treatment of mental disorders; (3) studies in any language other than English; and (4) studies published in 2022 because such studies may not have been completely published.

After searching according to the above criteria, 212 studies were downloaded from the WoS, and a total of 104 research articles remained after the screening. Information was extracted from each article, including title, authors and their e-mail addresses, year of publication, keywords, research field, article links, and citation count.

### 2.3. Research Framework

The research framework illustrates the analysis results based on scale, time, space, and composition. “Scale” includes descriptive and LDA analyses. “Time” refers to the analysis of the timeline of the 104 studies. “Space” refers to the distribution of research capacity analyzed by country. “Composition” refers to the network visualization analysis (Figure 2).

## 3. Results

This study discusses the research on intervention programs for smartphone addiction from four dimensions and explores the research trends of intervention programs in terms of scale, time, space, and composition.

### 3.1. Scale

In this section, “scale” refers to the number of published journal articles. This study selected 104 journal articles for analysis, including the total number of articles on intervention programs in each year from 2014 to 2021, topic correlation, research objects, and classification of intervention programs.

#### 3.1.1. Analysis of the Development of Number of Articles on Intervention Programs by Year

Using a descriptive analysis of studies from the WoS from 2014 to 2021, 104 studies were selected. First, it was noted that 2014 was the year that the earliest study on intervention programs was published and 2021 was the year with the highest number of studies, with a total of 38 studies (36.5%). Second, 2020 saw the publication of 27 articles (26%), indicating that scholars had increasingly valued the significance of intervention programs for smartphone addiction. Table 1 shows the overall increasing trend of research on intervention programs.

#### 3.1.2. Journal Article Citations and Percentages

A total of 104 articles published on the WoS by 74 research journals were selected. This study considered popular research journals as an analysis target. Four journals were identified to have been cited hundreds of times in this field: *Computers in Human Behavior* had the most citations (n = 253, 14.43%), followed by *Social Science Computer Review* (n = 238, 13.58%), *Children and Youth Services Review* (n = 204, 11.63%), and *Journal of Medical Systems* (n = 113, 6.45%).

#### 3.1.3. Research Topic/Abstract Analysis

The study applied LDA to the titles and abstracts to classify them into 10 research topics (Table 2). Among the publications, age group (children, adolescents, college students, adults) was the most commonly visited topic in studies on intervention programs for smartphone addiction (n = 90, 30.61%), followed by mental state (well-being, boredom, loneliness, fear of missing out, mental disorders, inattention) (n = 58, 19.73%). The third was usage behavioral problems (dependence, attachment, gaming, social media) (n = 51, 17.35%). The fourth was living conditions (stress, learning, society, sleep, family, bullying) (n = 45, 15.31%). The fifth was physical problems (obesity, bone problems, posture), the sixth was supervision and management system, and the seventh was personality factors (emotional imbalance), each one of which was visited at identical rates (n = 11, 3.74%). The eighth was gender (n = 8, 2.72%); the ninth was work (n = 5, 1.7%); and the tenth was COVID-19 (n = 4, 1.36%). These are recurring research themes in interventions for smartphone addiction. Additionally, most of the studies were related to multiple topics, such as age group and mental state, age group and usage behavioral problems, usage behavioral problems and living conditions, living conditions and physical problems, age group and gender, usage behavioral problems and gender, etc. This demonstrates that smartphone addiction is associated with a range of psychological, physical, and social factors. Understanding the influencing factors of specific subjects and tailoring intervention programs can improve their effectiveness.

#### 3.1.4. Research Subject Analysis

The LDA analysis showed that among 104 studies, students (children, adolescents, college students) (n = 59, 56.7%) were the most studied subjects (Table 3) since young people are globally recognized to be subjects with the highest daily usage rate of smartphones [46]. This has prompted scholars to conduct multi-faceted research on young people, focusing on psychological and physical disorders and life stress, as well as their relationships with schools and families, suggesting that social environments and interpersonal relationships both have an impact on their lives [47,48,49,50,51]. The second group was the general public (n = 34, 32.7%), followed by parents (n = 6, 5.77%), general workers (scholars, farmers, craftsmen, and merchants) (n = 4, 3.85%), and patients with mental disorders (n = 1, 0.96%).

#### 3.1.5. Analysis and Definitions of Intervention Program Classifications

This section applies the LDA analysis to the intervention program of each study, classifying 10 types of programs based on 104 studies. Table 4 “Psychological interventions” (n = 48, 26.97%) were the most commonly applied intervention method, indicating that smartphone addiction is associated with psychological factors. People vulnerable to depression, suicidality, and social disconnection may desire a sense of security by using their smartphones [50]. “Social support” interventions (n = 27, 15.17%) show that smartphone addicts need support from peers or the surrounding environment to overcome loneliness and to emerge from a “dark cloud” to ultimately establish good relationships with others and experience a sense of happiness [52,53]. “Lifestyle interventions” (n = 25, 14.04%) focus on regulating smartphone usage through restricting behaviors, such as limiting the time spent on gaming applications and prohibiting users from becoming “smartphone zombies” [54,55]. “Technological interventions” (n = 19, 10.67%) aim to create apps to remind smartphone users of their prolonged usage and response software for vehicle drivers who use smartphones, as well as to integrate medical treatments in the intervention of smartphone addiction [56,57,58]. “Family interventions” (n = 18, 10.11%) encourage family members to place restrictions on and influence addicts to help them feel cared for, thereby increasing family interactions [59]. “Medical interventions” (n = 14, 7.87%) concentrate on psychological and physical treatments for depression, anxiety, bad posture, and sleep problems [60,61]. “Educational intervention” programs (n = 12, 6.74%) use counseling sessions to teach children and adolescents to use smartphones correctly and organize psychosocial education courses [62,63]. “Exercise interventions” (n = 8, 4.49%) allow subjects to focus on exercise and reduce their temptation to use their phones, thus, promoting weight loss and health benefits [64,65]. “Mindfulness interventions” (n = 5, 2.8%) enhance self-belief, facilitate social adjustment, and reduce the occurrence of smartphone addiction [66,67]. “Meditation interventions” (n = 2, 1.12%) help exercise the body and mind and reduce smartphone obsession through seated meditation [23] (Table 4).

These studies on intervention programs may not only focus on one program but instead test the benefits of a combination of programs, for example, psychological and exercise, technological and medical, and lifestyle and educational with social support.

### 3.2. Time

In this section, “time” refers to the development trajectory of publications in different time periods. As shown in Figure 3, the earliest studies on intervention programs—“technological intervention,” “social support,” and “medical intervention”—emerged in 2014, based on analyses and screenings. In 2021, the direction of research on social support intervention shifted toward social anxiety, advocating social networking and relationships to maintain well-being, thereby reducing problematic smartphone use [62].

Studies on “psychological intervention” and “educational intervention” emerged in 2015. In 2015 and 2019, researchers proposed a combination of psychological and exercise interventions to reduce the occurrence of smartphone addiction using weight loss as a goal [65,68]. Studies on “lifestyle intervention” appeared in 2016, when scholars discussed the association between smartphone addiction intervention and gender differences in sleep quality [69].

In this research domain, “family intervention” and “mindfulness intervention” began receiving attention in 2018. Parental intervention in the use of smartphones among children and adolescents may improve well-being in the parent–child relationship and enable a desirable childhood environment [70,71,72]. Similarly, in 2018, studies increasingly showed the impact of a mindfulness intervention on smartphone addiction [73]. “Exercise intervention” was proposed in 2019 and integrated with psychological and family interventions. Additionally, studies in 2020 suggested that exercise can also enhance physical and mental states and offer early intervention for smartphone addiction [64,65].

Recently, a few academics have proposed “meditation intervention.” In 2020, researchers suggested that meditation can be applied to intervene in mental health to reduce smartphone addiction [23]. In particular, a 2021 study suggested that the impact of COVID-19 has led to a substantial rise in smartphone addiction and proposed mediation as a way to help alleviate smartphone addiction and fear of the pandemic [74].

During this period, it is observed that the research on intervention programs increasingly focused on non-medical methods as physical and mental treatments and relaxation activities available in society were innovated each year.

### 3.3. Space

“Space” refers to the international geographical distribution of academic research. China had the highest research volume in terms of articles on smartphone addiction published in international academic journals, followed by South Korea, Taiwan, Spain, the UK, the US, India, Canada, Hong Kong, Iran, Australia, Italy, Saudi Arabia, the Caucasus region, France, the Hebrew region, Lebanon, Tunisia, the Philippines, Switzerland, Nigeria, Germany, Bangladesh, Oman, Sudan, Austria, Palestine, Vienna, Germany, Switzerland, Peru, and Brazil (Figure 4). Information on the spatial distribution showed that stronger attention was placed on the issue in the East Asian region. In addition to Spain, the non-Asian countries mentioned above are mostly English-speaking countries. This is also possibly because the journal database mainly comprises studies published in English.

### 3.4. Composition

In this section, “composition” refers to understanding the structure of professional knowledge in a certain domain through exploring research topics and the types of relationships. Figure 5 illustrates the citation structure relationships among studies and indicates that the research of intervention programs for smartphone addiction is recognized by many academic journals and that they are interconnected in this field. *Computers in Human Behavior* has the highest number of citations. Similarly, the visualization shows that this journal has the strongest connections and mutual influence with other journals.

According to a VOSviewer analysis, the top-three researchers out of the 4935 in the total citations were the US clinical psychologist Jon D. Elhai, the Swiss experimental psychopathologist J. Billieux, and M. Kwon, who is a quantum optics scholar based in Boston (Table 5, Figure 6). The visualization analysis depicted in Figure 5 shows clusters around lead researchers in the field, indicating that their research has a high degree of association with others. The high concentration of researchers suggests cooperative relationships among multiple authors, as evidenced by the co-citations.

After analyzing 41 countries, China was found to have the highest number of citations in research on intervention programs for smartphone addiction, followed by South Korea, Taiwan, the US, the UK, Israel, Australia, Singapore, Spain, and Austria (Table 6). In Figure 7, denser dots and stronger connections indicate higher cooperation. China and South Korea not only showed high research capacity but also had highly influential and cooperative relationships.

Finally, a co-occurrence analysis of keywords was performed on the studies on intervention programs for smartphone addiction. According to the VOSviewer results, there were a total of 420 keywords. Smartphone addiction and problematic smartphone use appeared the most, totaling 62 occurrences (Table 7).

Figure 8 shows that certain keywords are intimately related to the intervention of smartphone addiction and represent related phenomena in that domain. Recently, researchers have proposed to replace “smartphone addiction” with “problematic smartphone use”. Most studies are related to mental health conditions such as depression and anxiety, and social media is also evidently related to smartphone addiction. Young people are the most common research subjects, and a few studies even suggest that an eating addiction among young people is related to smartphone addiction [74].

## 4. Discussion and Conclusions

This study extracted research studies available on the WoS and applied bibliometric methods to analyze the research trends of the intervention of smartphone addiction over the past eight years. The results support three conclusions. First, intervention programs can be classified into 10 types, namely psychological, social support, lifestyle, technological, family, medical, educational, exercise, mindfulness, and meditation. In particular, meditation is the latest addition, whereas psychological intervention has always been an essential intervention method for smartphone addiction, thereby indicating that addictions are significantly associated with psychological factors.

Second, based on the analysis results of 104 studies, the research volume of intervention programs has gradually increased, showing that researchers are becoming increasingly aware of the significance of smartphone addiction intervention. Additionally, during the search and analysis, we found that the number of studies on intervention programs for smartphone addiction, up to August 2022, was even greater than that of the year 2021. Owing to the increase in usage, smartphone addiction will become a more prominent issue and there will be more intervention programs in the future.

Third, from a geographical perspective, China and South Korea were found to be the countries with the most research on smartphone addiction intervention. Whether in terms of the number of publications or citations, both countries are the leaders and most influential in this research field. Moreover, the literature review showed that China, Saudi Arabia, and Malaysia were the countries with the highest population of smartphone addicts [8]. China produces the most research on interventions; however, Saudi Arabia and Malaysia, which have the second- and third-highest populations of smartphone addicts, produce very few studies on this domain.

Finally, by analyzing the most influential journals publishing articles on smartphone addiction, we identified that the researchers mainly had a human behavior or social science academic background. Additionally, we found that the influence and quality of journals are positively correlated. Studies have not regarded smartphone addiction as a disorder; instead, the phenomenon has generally been defined in terms of individual behaviors and social relationships. Yet, many studies have shown that smartphone addiction is related to physical and mental disorders. The most influential and cited researcher was Jon D. Elhai, a professor of clinical psychology at the University of Toledo. His research focuses on smartphone addiction and psychological issues such as anxiety, depression, and boredom.

This study explored the academic field of smartphone addiction intervention methods through data analysis, and the results serve as a significant reference for research in this domain. In addition to clarifying the key scholars, academic journals, and advancements of intervention programs, it provides insights into intervention in smartphone addiction worldwide. In light of this, we propose our recommendations for future research based on the results. First, in the analysis of each study, we found that qualitative and quantitative research methods accounted for the majority; however, the effectiveness of intervention methods requires further discussion. Second, this study conducted bibliometric analyses of research studies on the intervention of smartphone addiction using WoS and VOSviewer. Despite many significant findings, there may be differences in other databases, as each database has its own subset of literature available. Therefore, future studies may utilize other databases to widen the range of exploration. Finally, this study classified and analyzed the trend of research on intervention programs for smartphone addiction, which is an unprecedented attempt. Our analysis found that many studies relied on students as research subjects because of convenience, disregarding smartphone addiction in middle-aged and elderly groups. Hence, future studies may focus on a wider range of subjects to investigate the reasons behind the occurrence of smartphone addiction and expand the scope of research.

## 5. Research Limitations

One limitation of this study is that larger and more complete datasets are not available. As the authors are still in the exploratory stage, only the Web of Science database was retrieved, and because of the diverse nature of this database, it was included in the retrieval of the study. However, since the research articles submitted at the end of 2021 would only be published in full in 2022, they were retrieved on 30 June 2022, to make the data more complete. Moreover, this study only examines the trend of interventions for smartphone addiction research and does not provide additional knowledge.

The most accurate answer to the question of whether China is the most researched country is not yet available, although this study’s analysis and related research indicate that China is currently the country with the greatest interest in smartphone addiction intervention and the highest rate of addiction.

## 6. Future Directions and Outlook

In any future research, the authors will consider more comprehensive issues and hope to find different intervention options. Furthermore, they hope to improve their own research and consider the innovation and effectiveness of interventions for smartphone addiction. The discussion and analysis of this study may be premature, although if new findings emerge in time, they will be included in follow-up research. In addition, this study’s analysis drew on certain current research on interventions for smartphone addiction, and the related topics of intervention studies were also classified; however, the analysis of the relevance of various articles may suffer from some shortcomings, which will be extensively explored in the future.

## Figures and Tables

**Figure 1 ijerph-20-03840-f001:**
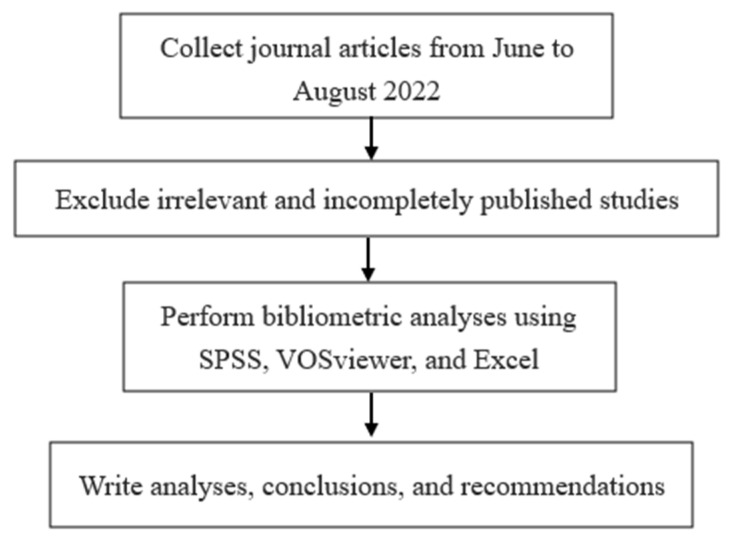
Research workflow.

**Figure 2 ijerph-20-03840-f002:**
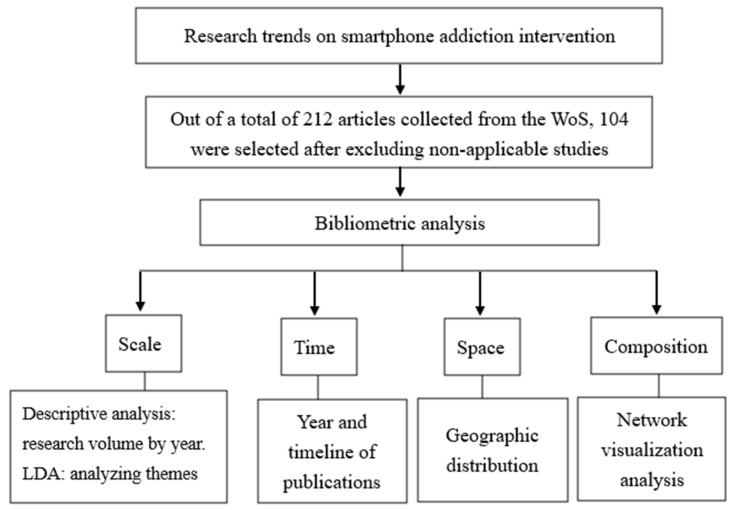
Research framework.

**Figure 3 ijerph-20-03840-f003:**
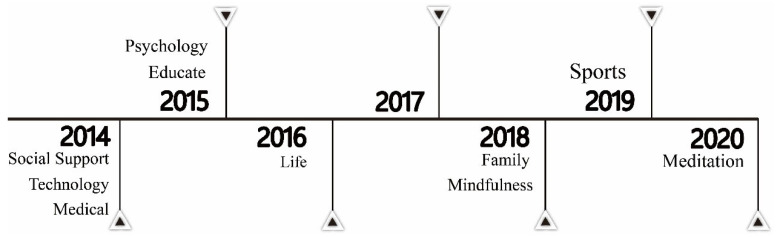
Emergence of each type of intervention program.

**Figure 4 ijerph-20-03840-f004:**
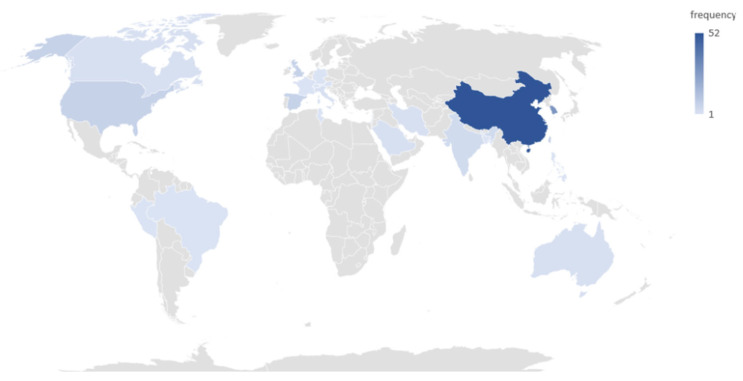
Countries/regions engaged in academic research.

**Figure 5 ijerph-20-03840-f005:**
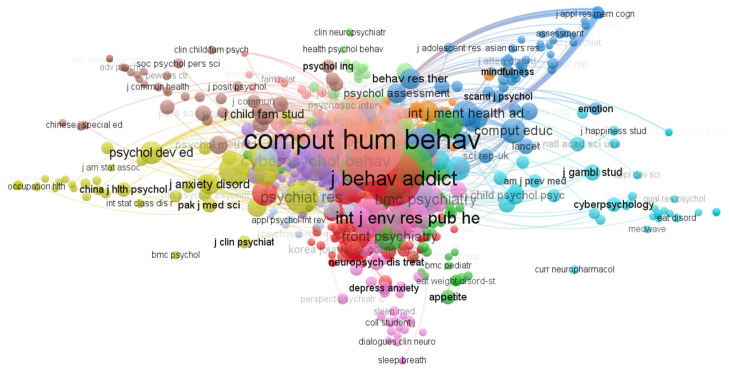
Citation analysis of source titles cited.

**Figure 6 ijerph-20-03840-f006:**
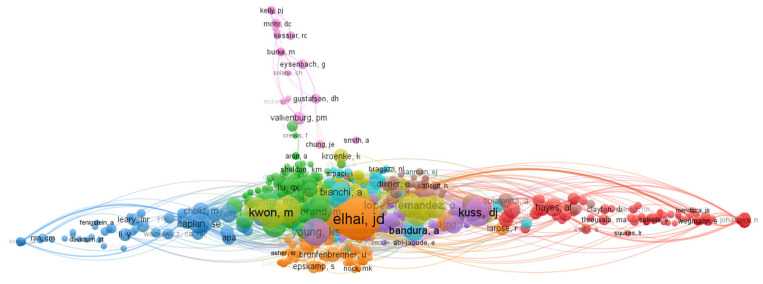
Co-citation analysis of cited authors.

**Figure 7 ijerph-20-03840-f007:**
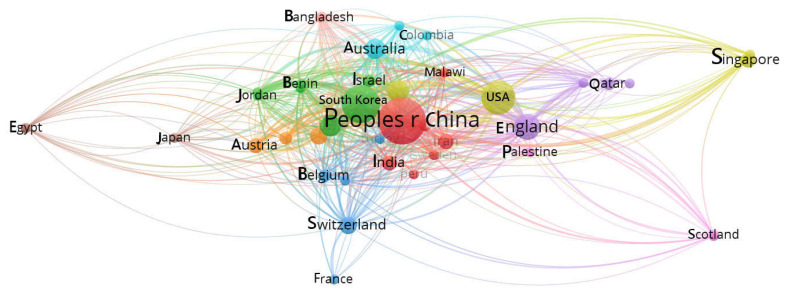
Bibliographic coupling of countries.

**Figure 8 ijerph-20-03840-f008:**
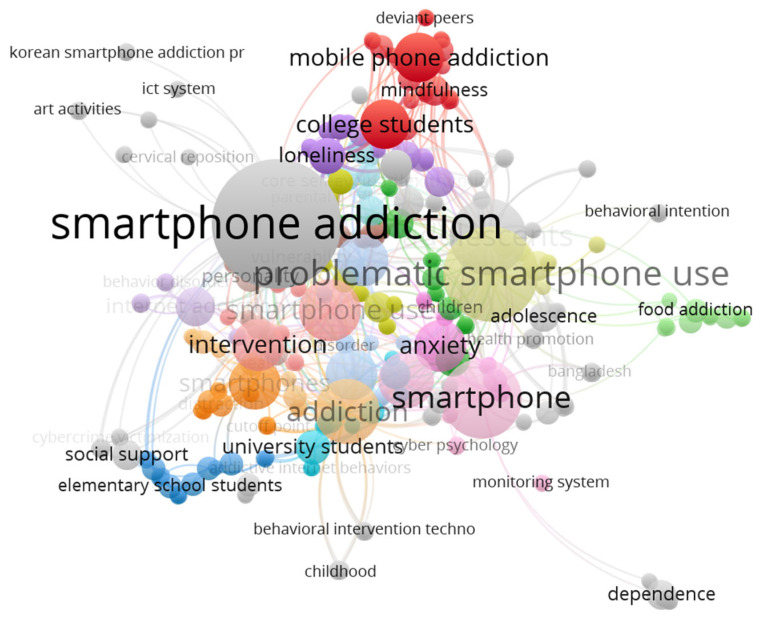
Co-occurrence of authors’ keywords.

**Table 1 ijerph-20-03840-t001:** Descriptive analysis (distribution of the total number of articles by year).

Year	N	Percentage
2014	3	2.9%
2015	2	1.9%
2016	4	3.8%
2017	6	5.8%
2018	13	12.5%
2019	11	10.6%
2020	27	26%
2021	38	36.5%
Total	104	100%

**Table 2 ijerph-20-03840-t002:** Ten research topics for the LDA analysis.

	Research Topics	N	Percentage
1	Age group (children, adolescents, college students, adults)	90	30.61%
2	Mental state (well-being, boredom, loneliness, fear of missing out, mental disorders, inattention)	58	19.73%
3	Usage behavior problems (dependence, attachment, gaming, social media)	51	17.35%
4	Living conditions (stress, learning, society, sleep, family, bullying)	45	15.31%
5	Physical problems (obesity, bone problems, posture)	11	3.74%
6	Supervision and management system	11	3.74%
7	Personality factors (emotional imbalance)	11	3.74%
8	Gender	8	2.72%
9	Work	5	1.70%
10	COVID-19	4	1.36%

**Table 3 ijerph-20-03840-t003:** Research subjects.

Subject	N	Percentage
Students (children, adolescents, college students)	59	56.7%
General public	34	32.7%
Parents	6	5.77%
General workers (scholars, farmers, craftsmen, merchants)	4	3.85%
Patients with mental disorders	1	0.96%

**Table 4 ijerph-20-03840-t004:** Intervention programs, definitions, frequencies, and percentages.

Intervention	Definition	N	Percentage
Psychological intervention	Smartphone addiction is related to depression and suicidality and addicts desire a sense of security by using their smartphones	48	26.97%
Social support	Smartphone addicts require support from peers or the surrounding environment	27	15.17%
Lifestyle intervention	Restrictions on smartphone usage in daily life	25	14.04%
Technological intervention	The usage of mobile phones is related to the development of apps	19	10.67%
Family intervention	Smartphone addicts increase family interactions through restrictions and influence placed upon by family members	18	10.11%
Medical intervention	Treatment associating smartphone addiction with psychological and physical problems	14	7.87%
Educational intervention	Smartphone addiction and psychosocial education courses	12	6.74%
Exercise intervention	Focus on exercise and reduce the temptation to use phones	8	4.49%
Mindfulness intervention	Enhancing self-belief and facilitating social adjustment	5	2.8%
Meditation intervention	Smartphone addiction and intervention through meditation	2	1.12%

**Table 5 ijerph-20-03840-t005:** Authors and total citations.

Ranking	Author	Total Citations	Ranking	Author	Total Citations
1	J. D. Elhai	80	6	K. Demirci	34
2	J. Billieux	62	7	O. Lopez-Fernandez	28
3	M. Kwon	40	7	D. Kardefelt-Winther	28
4	K. S. Young	36	9	A. J. A. M. van Deursen	27
5	D. J. Kuss	35	10	I. Leung	26

**Table 6 ijerph-20-03840-t006:** Countries and total citations.

Ranking	Country	Total Citations	Ranking	Country	Total Citations
1	China	645	6	Israel	77
2	South Korea	304	7	Australia	71
3	Taiwan	193	8	Singapore	54
4	USA	105	9	Spain	42
5	UK	80	9	Austria	42

**Table 7 ijerph-20-03840-t007:** Occurrence of keywords.

Ranking	Keyword	Occurrence	Ranking	Keyword	Occurrence
1	Smartphone addiction	39	7	Depression	8
2	Problematic smartphone use	23	8	Anxiety	8
3	Smartphone	17	8	Mental health	8
4	Adolescents	16	8	Intervention	8
5	Social media	9	9	College students	7

## Data Availability

Not applicable.

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
