# Peer review of "A Bibliometric Analysis to Identify Research Trends in Intervention Programs for Smartphone Addiction"

_ijerph, 2023, doi:10.3390/ijerph20053840_

Round 1

Reviewer 1 Report

I appreciate the authors’ efforts in the work put into the research of a bibliometric analysis to identify research trends in intervention programs for smartphone addiction:

To enhance the quality of the study, the authors must do some revision of their research and pay attention to several important issues:

(1)   The literature review section is missing. This should provide a good background of the topic and the authors must bring relevant and interesting arguments for their research based on other authors’ works.

(2)   The authors should explain why they limited their search between June 30, 2022, and August 31, 2022.

(3)   The research presented in the study has been carried out using Web of Science database. The authors should justify why was this used, instead of other recognised by scholars. In this direction, authors are advised to study and cite some updated works such as the following:

-      Bibliometric Analysis of the Green Deal Policies in the Food Chain. Amfiteatru Econ. 2022, 24, 410–428. DOI:10.24818/EA/2022/60/410.

-      Mapping Knowledge Area Analysis in E-Learning Systems Based on Cloud Computing. Electronics 2023, 12, 62. https://doi.org/10.3390/electronics12010062.

-      Exploring the Research Regarding Frugal Innovation and Business Sustainability through Bibliometric Analysis. Sustainability. 2022, 14(3), 1326. https://doi.org/10.3390/su14031326.

(4)   I recommend including a section of Findings, where authors could detail the research findings.

(5)   Authors need to explain about limitations and future research directions as well in revised draft.

(6)   The citation in the text and the reference list is not formatted according to the MDPI guidelines. Authors should correct this aspect.

Author Response

(1)   The literature review section is missing. This should provide a good background of the topic and the authors must bring relevant and interesting arguments for their research based on other authors’ works.

A. Interesting points about the literature analyzed in this study from 2014 to 2021 are already included in the text. If this is a question that I cannot answer right now, I will continue to study it and address it in follow-up research.

(2)   The authors should explain why they limited their search between June 30, 2022, and August 31, 2022.

A. I have written in Research limitations.

(3) The research presented in the study has been carried out using Web of Science database. The authors should justify why was this used, instead of other recognised by scholars. In this direction, authors are advised to study and cite some updated works such as the following:

  • Bibliometric Analysis of the Green Deal Policies in the Food Chain. Amfiteatru Econ202224, 410–428. DOI:10.24818/EA/2022/60/410.
  •      Mapping Knowledge Area Analysis in E-Learning Systems Based on Cloud Computing. Electronics 2023, 12, 62. https://doi.org/10.3390/electronics12010062.
  • Explore research on frugal innovation and business sustainability through bibliometric analysis. sustainability. 2022, 14(3), 1326. https://doi.org/10.3390/su14031326.

A. The reasons for using Web of Science databases are stated in the Research limitations section, but additional reasons are that the school has not purchased other analysis programs suitable for this database and personal funds are limited. I did not address these issues in the study limitations section because they are personal matters.

(4) I recommend including a Findings section where authors can elaborate on their findings.

A. Already covered, but research may be lacking at present. If found in the future, it will be included in follow-up research.

(5) The authors need to state the limitations and future research directions in the revised manuscript.

A. Added study limitations and future directions and outlook (6) Citations and reference lists in the text are not formatted according to MDPI guidelines. The authors should correct this aspect.

A. solved

Reviewer 2 Report

The article is a traditional bibliometric study. In this sense, it is not news. However, the topic discussed is of great interest. I mainly highlight that the increasing weight of Asia in scientific production and management is highlighted.

Author Response

Dear Reviewer 2
Thank you for your compliment

Reviewer 3 Report

The study was well described, but the presentation of the results has a very simple formula. The presentation (results) lacks even the simplest sets of correlations. In my opinion, even the simplest correlations should be sought, for example, to analyze why "China had the highest research volume in terms of articles on smartphone addiction pub-310 lished in international academic journals". Alternatively, cite relevant scientific research in this regard. In the current state of research, research does not add much to our knowledge, it only (partially) confirms the already widely described tendencies.

Another shortcoming of the article is the very limited part of the "discussion", in my opinion, the results should be related to other studies carried out.

In addition, in my opinion, figures 5, 6, 7, 8 are illegible, they should be reworked so that valuable information can be obtained from them.

Also in Figure 7, country names should be capitalized.

A relatively small additional effort can significantly increase the scientific value of the text.

Author Response

1.The study was well described, but the presentation of the results has a very simple formula. The presentation (results) lacks even the simplest sets of correlations. In my opinion, even the simplest correlations should be sought, for example, to analyze why "China had the highest research volume in terms of articles on smartphone addiction pub-310 lished in international academic journals". Alternatively, cite relevant scientific research in this regard. In the current state of research, research does not add much to our knowledge, it only (partially) confirms the already widely described tendencies.

A. This study’s analysis drew on certain current research on interventions for smartphone addiction, and the related topics of intervention studies were also classified, but the analysis of the relevance of various articles may suffer from certain shortcomings, which will be extensively explored in the future. The most accurate answer to the question of whether China is the country with the highest research volume is not yet available, but the analysis of this study and related research indicates that China is currently the country with the highest interest in smartphone addiction intervention and the highest rate of addiction.

2.Another shortcoming of the article is the very limited part of the "discussion", in my opinion, the results should be related to other studies carried out.

A. The discussion and analysis of this study may be premature, but if new findings emerge in time, they will be included in follow-up research.

3.In addition, in my opinion, figures 5, 6, 7, 8 are illegible, they should be reworked so that valuable information can be obtained from them.

A. Problem has been solved

4.Also in Figure 7, country names should be capitalized.

A. Problem has been solved